# Exploiting Semantic Relations for
# Fine-grained Entity Typing

**Hongliang Dai**                                          HDAI@CSE.UST.HK
*Department of Computer Science and Engineering, HKUST, Hong Kong, China*

**Yangqiu Song**                                          YQSONG@CSE.UST.HK
*Department of Computer Science and Engineering, HKUST, Hong Kong, China*
*Peng Cheng Laboratory, Shenzhen, China*

**Xin Li**                                                ALONSOLI@TENCENT.COM
*Tencent Technology (SZ) Co., Ltd., Shenzhen, China*

## Abstract

Fine-grained entity typing results can serve as important information for entities while constructing knowledge bases. It is a challenging task due to the use of large tag sets and the requirement of understanding the context. We find that, in some cases, existing neural fine-grained entity typing models may ignore the semantic information in the context that is important for typing. To address this problem, we propose to exploit semantic relations extracted from the sentence to improve the use of context. The used semantic relations are mainly those that are between the mention and the other words or phrases in the sentence. We investigate the use of two types of semantic relations: hypernym relation, and verb-argument relation. Our approach combine the predictions made based on different semantic relations and the predictions of a base neural model to produce the final results. We conduct experiments on two commonly used datasets: FIGER (GOLD) and BBN. Our approach achieves at least 2% absolute strict accuracy improvement on both datasets compared with a strong BERT based model.

## 1. Introduction

Given a sentence and an entity mention in it, fine-grained entity typing (FET) aims to recognize the fine-grained types of this mention that matches the context [Ling and Weld, 2012]. For example, in the sentence "Trump's trade wars and ethanol policy hurt farmers," the entity mention "Trump" should be identified as a politician. Other types of Donald Trump such as businessman, TV personality are incorrect since they do not fit the context. FET provides more information for the entity mentions to downstream tasks than the coarse-grained typing used in named entity recognition [Borthwick et al., 1998]. Its results can be used as important information for the referred entities while constructing knowledge bases. It can also be applied to relation extraction [Koch et al., 2014], question answering [Lee et al., 2006], entity linking [Ling and Weld, 2012], etc.

Though there have been many studies on FET, the neural models used to approach this problem are usually similar. Typically, they [Shimaoka et al., 2017, Onoe and Durrett, 2019, Dai et al., 2019] first obtain two representations from the mention and the context, respectively. Then, they concatenate them to form a feature representation that is used for the final prediction. While obtaining the representation for the context, all words are uniformly treated as the input for a neural model such as LSTM. We find that such models

| |
|---|
| **Sentence**: Connaught is a biotechnology research and vaccine manufacturing concern. |
| **Gold**: /ORGANIZATION, /ORGANIZATION/CORPORATION |
| **Predict**: /ORGANIZATION |
| **Sentence**: Characters drink Salty Dogs, whistle "Johnny B. Goode" and watch Bugs Bunny reruns. |
| **Gold**: /SUBSTANCE, /SUBSTANCE/FOOD |
| **Predict**: /WORK_OF_ART |

Table 1: Examples of the predictions made by the model used in [Dai et al., 2019] on the BBN dataset. Entity mentions are underlined.

may overlook some semantic relations between the mention and the words or phrases in the sentence that are important for determining the entity type.

To illustrate this problem, we show two examples in Table 1, along with the predictions made by the state-of-the-art FET model used in [Dai et al., 2019], which is a deep neural model that follows the typical architecture described above. It achieves a strict accuracy of 0.805 on the BBN dataset [Weischedel and Brunstein, 2005]. In the first example, although the sentence clearly says that "Connaught" is a "concern" (company), the model only predicts it as /ORGANIZATION. This problem can be resolved if we can extract "concern" as the hypernym of "Connaught", and correspond it to the type /ORGANIZATION/CORPORATION. The hypernyms of mentions occur frequently when the referred entities are newly occurred or not known by a lot of people, which makes the typing results based on them especially valuable. In the second example, the model incorrectly predicts the mention "Salty Dogs" as /WORK_OF_ART, likely because of the terms in the context such as "Characters", "Bugs Bunny". However, since "Salty Dogs" is the *patient* of the verb "drink", we know that the type of this mention should probably be /SUBSTANCE/FOOD (drinks are categorized as /SUBSTANCE/FOOD in BBN). Such verb-argument relations can be obtained through Semantic Role Labeling (SRL) [Palmer et al., 2005]. Some other types of semantic relations can also help. For example, from the relation between "head coach" and "Stanford" in "the head coach of Stanford", we can infer that this "Stanford" probably refers to a sports team.

Thus, we believe that it is possible to improve neural FET models by using some semantic relations between the mention and the context words or phrases. In this paper, we investigate how the two types of semantic relations that is already discussed above can be exploited in FET: hypernym relations, and the verb-argument relations obtained with semantic role labeling. To utilize them effectively, we design two individual components that performs FET based on these two types of relations respectively.

Since such semantic relations can only be found for some mentions, an existing neural FET model is also used as a base model. Then, we use a stacking approach similar to mixture of experts [Jacobs et al., 1991] to combine these three components, so that the final results are better than the predictions made by each individual component. Such a design makes our approach very flexible. It allows other types of semantic relations to be incorporated easily, and has little restriction on how each of them are modeled.

We conduct experiments on two commonly used FET datasets: FIGER (GOLD) [Ling and Weld, 2012] and BBN. Experimental results show that our approach effectively improves FET performance. Both hypernym relations and verb-argument relations are helpful.

Our main contributions are:

- We are the first to propose using the semantic relations between the mention and some words or phrases in the sentence to improve fine-grained entity typing. A framework that can incorporate different types of such relations is proposed.

- We investigate the use of hypernym relations and verb-argument relations for FET.

- We conduct experiments on two commonly used FET datasets to verify the effectiveness of our approach.

Our code is available at `https://github.com/HKUST-KnowComp/SRFET`.

## 2. Related Work

Most recent studies on FET use neural models to approach this problem. They try different ways to improve the performance. For example, denoising the training data [Onoe and Durrett, 2019, Chen et al., 2019, Ren et al., 2016b], utilizing the inter-dependencies between different types [Lin and Ji, 2019, Xu and Barbosa, 2018], using external information [Xin et al., 2018, Dai et al., 2019], etc.

Before neural models become popular, there were also some approaches [Yogatama et al., 2015, Ling and Weld, 2012, Ren et al., 2016a] that use manually designed features. However, they usually use lexical and syntactic features instead of semantic features. The performance of these approaches is also inferior than the neural FET models proposed more recently.

There are several studies [Choi et al., 2018, Yogatama et al., 2015, Ren et al., 2016a] that make use of the syntactic head of the mention strings. They use the head words as features or try to create more training data by mapping them to the types used by the dataset. This is different from the hypernym relation based FET used in our approach. Head words are only extracted from the mention strings, and they do not necessarily indicate the fine-grained types of the mentions. But we extract hypernyms from the whole sentence to help FET. They are almost always the terms that describe the types of the mentions.

Our work is also related to using semantic annotation results to improve NLP tasks. For example, semantic role labeling have been used to help question answering [Hommel et al., 2019, Khashabi et al., 2018], machine translation [Marcheggiani et al., 2018], etc.

## 3. Framework

The framework of our approach is shown in Figure 1. Firstly, there are three individual components that makes FET predictions: *Base FET Model*, *Hypernym Relation Based FET*, and *Verb-argument Relation Based FET*. For each mention, each component outputs a vector whose dimensionality equals to the number of fine-grained type tags. Each element in the vector corresponds to a type, with positive value indicates that the mention belongs to this type, negative value indicates that the mention does not belong to this type. The Base FET Model is a neural model with a structure similar to existing FET models. It

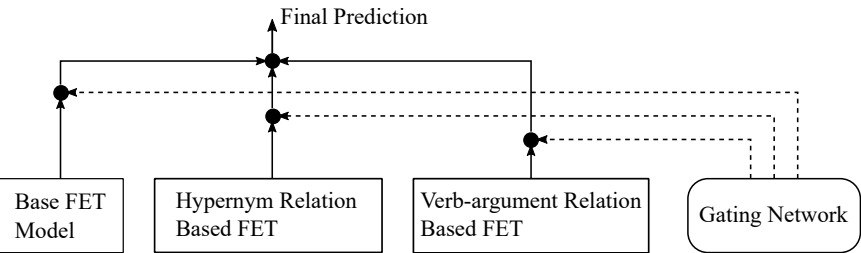

Figure 1: Framework of our approach.

simply takes the mention and its context sentence as input, and outputs a prediction result. The other two components are only active when the corresponding semantic relation is found for the mention. They output zero vectors when inactive.

We use a gating network [Jacobs et al., 1991] to combine these three components. It takes some features we extract from the three components as input, and outputs three weights. Then, they are used to take a weighted sum of the output vectors of the three components to obtain the final prediction.

Next, we will introduce Hypernym Relation Based FET in Section 4, Verb-argument Relation Based FET in Section 5. The gating network will be described in Section 6.

## 4. Hypernym Relation Based FET

There are many existing studies for hypernym detection. However, almost all of them focus on classifying whether a pair of words are in hypernym relation after processing a large textual corpus [Camacho-Collados et al., 2018]. It is common for them to use statistical information that is obtained based on all the co-occurrences of this word pair in the corpus. This does not satisfy our requirement, since we need an approach that extracts the hypernym of a mention only from the sentence it occurs in, instead of using the whole corpus. Also, we need the extracted hypernyms to be of high precision, since incorrect hypernyms will hurt the performance of our FET system. Moreover, we need to be able to map the extracted hypernyms to the type labels used by the FET datasets. To this end, we propose a two-step hypernym extraction approach:

1) **Hypernym Candidate Extraction**. For each mention, use rules or string matching to extract words or phrases from the sentence that are possible to be its hypernyms. We call these words or phrases hypernym candidates.

2) **Hypernym Verification**. For each hypernym candidate found in step 1, we use a neural model to predict whether it truly is a hypernym of the mention.

### 4.1 Hypernym Candidate Extraction

To perform hypernym candidate extraction, we first construct a dictionary of type terms by using the Wikidata knowledge base. In Wikidata, each entity or concept is represented as an item. Each entity item may have an attribute called *instance_of*. For example, the entity *Harvard University* has an *instance_of* value *private university*, the entity *Google* has an *instance_of* value *enterprise*. Items such as *private university* and *enterprise* may have

an attribute called *subclass_of*. For example, *private university* has an *subclass_of* value *university*, *enterprise* has an *subclass_of* value *organization*. We process a Wikidata dump and find the items that appear as *instance_of* values or *subclass_of* values of some other items as type terms.

We also use Wikidata to construct a mapping from the type terms obtained above to the type tags used by the target dataset. For each type term $s$, we perform a depth-limited breadth first search starting from the its corresponding Wikidata item, along the *subclass_of* attribute. Denote all the visited items as $H(s)$. $H(s)$ contains types or concepts that are more general than $s$. Then, make a series of attempts to map $s$ to the target tags based on $H(s)$. Bellow is a snippet of the process:

IF $\langle Q6256, country \rangle \in H(s) \cup \{s\}$ THEN
    RETURN */GPE/COUNTRY*
IF $\langle Q6266, nation \rangle \in H(s) \cup \{s\}$ THEN
    RETURN */GPE/COUNTRY*
IF $\langle Q515, city \rangle \in H(s) \cup \{s\}$ THEN
    RETURN */GPE/CITY*
...

Where $\langle Q6256, country \rangle$ represents the item in Wikidata with ID "Q6256" and label "country". The rest of the process is similar, and can be obtained with the help of an FET training set. The type terms that cannot be mapped are removed from the dictionary. We denote the final set of type terms in the dictionary as $S$, and the resultant mapping as $\tau$. For example, for term "city" $\in S$, $\tau(\text{"city"}) = /\text{GPE/CITY}$. Note that $\tau$ is only used for hypernym extraction and verification, it maps a term to one type tag. While performing FET based on the extracted term, a slightly different mapping is used, this will be discussed later.

After obtaining the type term set $S$ and the mapping $\tau$. For each mention, we extract term $s \in S$ as its candidate hypernym if it occurs in the context sentence, and matches any one of a list of patterns we create. The full list of patterns can be found in Appendix A. Below are three of them:

- $M$ (and|or) (any|some) other $s$. E.g., "Microsoft and other companies".

- $M$ (is|,) (a|an) (*NN*|*NNS*|*NNP*|*JJ* |*CC* |,)* $s$. E.g., "Connaught is a biotechnology research company".

- $s$ $M$. E.g., "actor Leonardo DiCaprio ...".

where $M$ denotes the mention string; *NN*, *NNS*, *NNP*, *JJ*, and *CC* matches words with the corresponding POS tags. The first pattern above is from Hearst Patterns [Hearst, 1992, Roller et al., 2018]. Besides these patterns, we also extract the terms in $S$ that occurs within the mention string itself. For example, we extract the term "university" from "Harvard University".

## 4.2 Hypernym Verification

The hypernym candidates extracted above can be inaccurate. Thus, we also design a BERT [Devlin et al., 2019] based neural model to verify them through binary classification.

We form the sequence "[CLS] sentence [SEP] mention string [SEP]" as the input of the BERT model. Then, we use the output vector corresponds to the first token "[CLS]" as a representation for the context and the mention. Denote this vector as $\boldsymbol{u}$. $\boldsymbol{u}$ is used as the input of an MLP with two fully-connected layers to obtain $\boldsymbol{v}$. We also assign an embedding vector to each type tag used by the dataset. Let $\boldsymbol{t}$ be the embedding vector corresponds to $\tau(s)$, the type tag obtained by mapping the extracted type term. Then, we calculate the score of whether $s$ is a hypernym of the current mention $m$ with $f(s,m) = \boldsymbol{v} \cdot \boldsymbol{t}$. Thus, our intention is to transform the binary classification problem of whether the extracted term is a hypernym of the mention into another binary classification problem of whether the mention belongs to the fine-grained type mapped through the extracted term. This simplifies the problem, since the type tag set used by the dataset is much smaller than the type term set $S$ we obtain from Wikidata. The type term $s$ is sometimes more specific than its mapped type tag $\tau(s)$. For example, "private university" is mapped to /ORGANIZATION/EDUCATIONAL. However, this usually does not affect the correctness of the prediction. We use the following loss to fine-tune the BERT model and train the other parameters:

$$L = \sum_m \max(0, 1 - y \cdot f(s(m), m)), \tag{1}$$

where we use $s(m)$ to denote the hypernym candidate extracted for mention $m$, $y$ equals to 1 when $s(m)$ is a true hypernym and -1 otherwise. With this loss, we regard the extracted hypernym candidates whose scores are larger than 0 as true hypernyms. If more than one hypernyms are found for a mention, we use the one with the largest score.

We use distant supervision to train our hypernym verification model. The training data is automatically generated by using the anchor links in Wikipedia. Similar to [Ling and Weld, 2012], each anchor link is treated as an entity mention, the Wikipedia page it points to is viewed as the entity this mention refers to. We run the hypernym candidate extraction step to obtain the possible hypernyms of these mentions. For each mention, if a term $s$ is successfully extracted as its hypernym candidate, we check whether $\tau(s)$ matches the type of the entity it refers to. We use the extracted term $s$ and this mention as a positive sample if $\tau(s)$ matches the entity type. Otherwise, they are used as a negative sample. The fine-grained types of the referred entity is obtained by using the *instance_of* and *subclass_of* values of its corresponding item in Wikidata, and following the process described in 4.1.

The verification step greatly helps to improve the precision of the extracted hypernyms. In our experiments, it improves FET prediction strict accuracy of the *Hypernym Relation Based FET* component from 0.55 to 0.75 on FIGER (GOLD), and from 0.79 to 0.94 on BBN.

Finally, if a term $s$ is detected as a hypernym for mention $m$ with our two step approach, the Hypernym Relation based FET component assigns $\tau'(s)$ as the fine-grained type labels of $m$. Here, $\tau'$ is slightly different from $\tau$ in that it may map a term to multiple type tags, since FET can be formed as a multi-class multi-label classification problem. For example, for the FIGER (GOLD) dataset, $\tau'(\text{"government agency"}) = /organization, /government\_agency$. $\tau'$ can also be obtained with the process described in Section 4.1. To represent this prediction in the form mentioned in Section 3, we construct a vector $\boldsymbol{h}_m \in \mathbb{R}^K$, where $K$ is the total number of type tags used by the target dataset. The $i$th element in $\boldsymbol{h}_m$ equals to 1 if the $i$th type is in $\tau'(s)$ and -1 otherwise.

## 5. Verb-argument Relation Based FET

We use SRL to obtain verb-argument relations. SRL detects the semantic arguments of the verbs in sentences and assign role labels to them [Carreras and Màrquez, 2004]. In this paper, we use an SRL system provided by AllenNLP[1], which implements the approach proposed in [Shi and Lin, 2019]. This approach adopts the PropBank [Palmer et al., 2005] convention of SRL annotation. It assigns numbers to the arguments of the verbs, e.g., Arg0, Arg1, Arg2, etc. Arg0 represents the *Agent*; Arg1 represents the *Patient*. Other arguments may have different meanings for different verbs. We only use Arg0, Arg1 and Arg2 since they occur most frequently [Palmer et al., 2005, Carreras and Màrquez, 2004]. Thus, we perform Verb-argument Relation Based FET only if a mention matches the Arg0, Arg1 or Arg2 of a verb in the sentence. When a mention matches an argument of a verb, we can sometimes infer its type from this verb or the combination of this verb and its other arguments. For example, consider the second sample in Table 1 again, the SRL system will annotate "Characters" and "Salty Dogs" as the Arg0 and Arg1 arguments of the verb "drink" respectively. In this case, the mention matches Arg1. Since the Arg1 (Patient) argument of the verb "drink" is usually a drink (corresponds to /SUBSTANCE/FOOD in BBN), this result helps to determine the type of the mention.

We use three neural models with the same structures for mentions that match Arg0, Arg1, and Arg2, respectively. Next, we only describe the model for mentions that match Arg0 in detail. Let $m$ be a mention that matches the Arg0 of a verb in the containing sentence. We obtain the following four vectors: 1) the average of the embeddings of the words in the mention string; 2) the embedding vector of the verb; 3) the output vector of the last time step of an LSTM applied to the word embedding sequence of Arg1; 4) the output vector of the last time step of another LSTM applied to the word embedding sequence of Arg2. For 3) and 4), we use a zero vector if the corresponding argument does not exist. These four vectors are concatenated and fed to an MLP with two fully connected layers to obtain a vector representation $\boldsymbol{x} \in \mathbb{R}^d$. Then, following [Dai et al., 2019], we create an embedding vector for each fine-grained type tag used by the target dataset, and use them to form a matrix $\boldsymbol{U} \in \mathbb{R}^{K \times d}$, where $K$ is the number of fine-grained type tags. Finally, we use $\boldsymbol{g}_m = \boldsymbol{U}\boldsymbol{x}$ as the output vector of the Verb-argument Relation Based FET component for mention $m$. We expect the $i$th element in the $K$ demential vector $\boldsymbol{g}_m$ to corresponds to the score of whether mention $m$ belongs to the $i$th type. To this end, the following loss is used to train the parameters:

$$L = \sum_m [\sum_{i=1}^{K} \max(0, 1 - y(m,i) \cdot \boldsymbol{g}_{mi})] \tag{2}$$

where $y(m,i)$ equals to 1 if $m$ belongs to the $i$th type and -1 otherwise.

## 6. Component Stacking

So far, for a mention $m$, we have a Hypernym Relation FET component, which outputs vector $\boldsymbol{h}_m$, and an Verb-argument Relation Based FET component, which outputs vector

---

1. https://demo.allennlp.org/semantic-role-labeling

$g_m$. As previously mentioned in Section 3, besides them, we also use a base neural model. Different models can be used as the base model, but its output has to be in the same form as the vector $g_m$ in Section 5. That is, it needs to output a vector whose $i$th element is the score corresponds to the $i$th type. We denote the output vector of the base model as $p_m$. Different base models are tried in our experiments, the details are in Section 7.

We use a gating network to combine $h_m$, $g_m$ and $p_m$. The input to this gating network are features extracted from the three components. Specifically, the following features are concatenated to form the input feature vector: the two largest elements in $g_m$; the two largest elements in $p_m$; the score output by the hypernym verification model. This feature vector is then used as the input of an MLP with two fully connected layers to obtain a three dimensional vector $u$. Denote $w_i$ as the $i$th element in softmax($u$). Finally, we obtain

$$\hat{y}_m = w_1 h_m + w_2 g_m + w_3 p_m. \tag{3}$$

We predict that $m$ belongs to a type if its corresponding element in $\hat{y}_m$ is greater than 0. The parameters are trained with a loss that has the same form as Equation 2.

With this method to ensemble the predictions of different components, we can easily apply more types of semantic relations to our framework.

## 7. Experiments

We conduct our experiments on two commonly used datasets: FIGER (GOLD) [Ling and Weld, 2012] and BBN [Weischedel and Brunstein, 2005]. FIGER (GOLD) contains 563 test samples manually labeled based on a tag set of 113 types; BBN contains 13,282 test samples labeled based on a tag set of 56 types. For training, we use the same training data used in [Dai et al., 2019], since their model is the state of the art, and is applied as the base model in our experiments. The training data is automatically constructed by using the anchor links in Wikipedia [Ling and Weld, 2012].

For each dataset, we manually annotated 500 mentions to train the gating network. For FIGER (GOLD), these mentions are randomly sampled from Wikipedia anchor links. For BBN, they are randomly sampled from its training set. They are further divided to form a training set that contains 400 mentions, and a development set that contains 100 mentions. For our approach, we run the ensemble part 10 times with different random initialization and report the average performance.

For pretrained BERT, we use BERT-Base cased. For the Verb-argument Relation Based FET models, the number of hidden units in the MLP and the type embedding size are both set to 500. For BERT (MASK) and BERT (SEP) in Section 7.1, the number of hidden units in the MLP and the type embedding size are both set to 600. While training BERT related base models, the batch sizes are all set to 32, the learning rate is set to 1e-5. For the gating network, the number of hidden units in the MLP is set to 5. While training it, we use L2 regularization, with the regularization parameter set to 0.1.

Three evaluation metrics are used: strict accuracy, Macro F1, and Micro F1 [Ling and Weld, 2012].

| Dataset | FIGER (GOLD) | | | BBN | | |
|---|---|---|---|---|---|---|
| Approach | Acc | Macro F1 | Micro F1 | Acc | Macro F1 | Micro F1 |
| CLSC | - | - | - | 74.7 | 80.8 | 80.5 |
| AMLTR | 62.9 | 83.0 | 79.8 | 55.9 | 79.3 | 78.1 |
| IFETEL (NoEL) | 69.8 | 82.7 | 80.4 | 80.5 | 87.5 | 88.0 |
| BERT (MASK) | 70.87 | 86.06 | 82.84 | 80.83 | 87.59 | 87.96 |
| BERT (SEP) | 69.80 | 84.64 | 81.67 | 84.31 | 89.09 | 89.62 |
| Ours | **74.21** | **86.77** | **84.17** | **86.52** | **90.62** | **90.99** |

Table 2: FET performance. Our approach uses BERT (MASK) as the base model on FIGER (GOLD), BERT (SEP) as the base model on BBN. "Acc" denotes strict accuracy.

### 7.1 FET Performance

To evaluate the performance of our approach, we compare with the following methods: **CLSC** [Chen et al., 2019] uses compact latent space clustering to improve the use of imperfect annotations. **AMLTR** [Lin and Ji, 2019] uses a mention-aware attention mechanism to let the model focus on important words in mentions and contexts. It also uses latent type representation to capture the inter-dependencies between types. **IFETEL** [Dai et al., 2019] is a state-of-the-art FET approach. It first perform entity linking to the mentions, then use the type information of the target entity to improve FET performance. **BERT (MASK)** and **BERT (SEP)** are two BERT [Devlin et al., 2019] based models whose structures follow the one used in IFETEL, but with the entity linking part removed. BERT (MASK) replaces the mention string with the "[MASK]" token, then feeds the sentence to BERT. The output vector corresponds to "[MASK]" is used as the context representation. BERT (SEP) feeds "[CLS] sentence [SEP] mention string [SEP]" to BERT, and use the output vector corresponds to "[CLS]" to replace both the mention representation and the context representation. For pretrained BERT model, we use the BERT-Base Cased version.

We first evaluate how our approach may improve the FET performance when Entity Linking is not used. The results are listed in Table 2. We can see that our approach effectively improves the performance on both datasets, even though the base model that uses BERT is already strong. The improvement is more significant on FIGER (GOLD) than on BBN. This is because BBN is relatively simpler. It uses a smaller tag set, and only assigns one tag to each mention. We also notice that the improvement is larger on strict accuracy than on Macro F1 and Micro F1. The base models already achieve high Macro F1 and Micro F1, which makes it hard to further improve these two measures. Achieving higher strict accuracy is important, for many mentions, it means that fine-grained types, instead of their corresponding super-types, are correctly predicted. This is because partially correct predictions (e.g., only predicting */organization* for a mention that is labeled as */organization*, */organization/company*) also contributes to Macro F1 and Micro F1, but are simply regarded as incorrect while using strict accuracy.

| Dataset | FIGER (GOLD) | | | BBN | | |
|---------|-----|----------|----------|-----|----------|----------|
| Approach | Acc | Macro F1 | Micro F1 | Acc | Macro F1 | Micro F1 |
| IFETEL | 75.5 | 87.1 | 84.6 | 82.5 | 89.2 | 89.6 |
| BERT+EL | 76.55 | 88.79 | 85.66 | 85.56 | 90.66 | 90.93 |
| Ours | **78.77** | **89.09** | **86.63** | **87.04** | **91.47** | **91.70** |

Table 3: FET performance when entity linking results are used. Our approach uses BERT+EL as the base model. "Acc" denotes strict accuracy.

| Dataset | FIGER (GOLD) | | | BBN | | |
|---------|-----|----------|----------|-----|----------|----------|
| Approach | Acc | Macro F1 | Micro F1 | Acc | Macro F1 | Micro 1 |
| Base Model | 70.87 | 86.06 | 82.84 | 84.31 | 89.09 | 89.62 |
| Base+HR | 72.84 | 86.39 | 83.74 | 86.26 | 90.49 | 90.83 |
| Base+VR | 72.15 | 86.36 | 83.26 | 84.69 | 89.33 | 89.86 |
| Full (No Ensemble) | 73.71 | **86.98** | **84.32** | 84.97 | 89.60 | 90.09 |
| Full | **74.21** | 86.77 | 84.17 | **86.52** | **90.62** | **90.99** |

Table 4: FET performance of different variants. "Base+HR" only uses hypernym relations. "Base+VR" only uses verb-argument relations. "Full (No Ensemble)" removes the gating network, and simply takes the average of all the predictions.

[Dai et al., 2019] finds that using the type information about the mentioned entity obtained through entity linking may greatly improve FET performance. Although this limits the applications of the final FET system (e.g., this makes it hard for the system to be applied to knowledge base construction), we also conduct experiments to see how our approach performs when the base model uses entity linking. The results are in Table 3. Besides the original model proposed in [Dai et al., 2019], we also report the results of improving it by replacing the BiLSTM part with pretrained BERT. Specifically, we use BERT (MASK) for FIGER (GOLD), BERT (SEP) for BBN. The results show that we can still improve the final performance by incorporating semantic relation information, though the improvement is less than when entity linking is not used.

## 7.2 Ablation Study

For ablation study, we first investigate how hypernym relations and verb-argument relations affects the final performance respectively. Thus, we evaluate the performance of our approach when only Hypernym Relation Based FET or only Verb-argument Relation Based FET is combined with the base model. The results are shown in Table 4. We can see that both hypernym relations and verb-argument relations are beneficial for FET. Hypernym relations are more helpful. This is mainly because predicting the fine-grained type of a mention based on the hypernym extraction results is more accurate. Moreover, the hypernyms extracted by our approach are of high precision.

| Dataset | FIGER (GOLD) | BBN |
|---|---|---|
| Mentions with HR (%) | 14.92 | 31.48 |
| Mentions with VR (%) | 34.64 | 26.06 |

Table 5: Semantic relation statistics.

| Dataset | FIGER (GOLD) | | | BBN | | |
|---|---|---|---|---|---|---|
| Component | Acc | Macro F1 | Micro F1 | Acc | Macro F1 | Micro 1 |
| Base Model | 70.87 | 86.06 | 82.84 | 84.31 | 89.09 | 89.62 |
| HR Based FET | 75.00 | 88.79 | 87.25 | 93.67 | 96.05 | 95.66 |
| VR Based FET | 69.23 | 84.30 | 80.63 | 78.94 | 83.29 | 83.85 |

Table 6: FET results of individual components. HR (Hypernym Relation) Based FET and VR (Verb-argument Relation) Based FET are evaluated only on the mentions they are active.

We also evaluate the effectiveness of the ensemble part of our approach. In Table 4, "Full (No Ensemble)" removes the gating network, and simply takes the average of the vectors $\boldsymbol{h}_m$, $\boldsymbol{g}_m$ and $\boldsymbol{p}_m$ described in Section 6. The performance of our full approach is similar with this variant on FIGER (GOLD), but is much better on BBN. Thus, the ensemble part is helpful for achieving stable good performance.

### 7.3 Semantic Relation Statistics

Table 5 reports some statistics about the semantic relations found in the two datasets we used. The amount of mentions that have hypernyms in BBN is more than we expected. There are two main reasons: 1) Many mentions are found to contain a hypernym inside its mention string. For example, the "university" inside "Harvard University" will be identified as a hypernym by our approach. 2) BBN contains some nominal mentions, which are themselves regarded as type terms by our approach and are extracted as hypernyms. For example, mentions such as "drug", "disease", etc. However, there are no such mentions in FIGER (GOLD). We do not discard such "hypernyms" since mapping them directly to the types used by the dataset also help to slightly improve the final performance.

Within the mentions that has verb-argument relations, the percentage that match the Arg0, Arg1, and Arg2 arguments are 74.4%, 21.5%, 4.1% on FIGER (GOLD), and 67.9%, 25.9%, 6.2% on BBN, respectively.

### 7.4 Individual Component Performance

We report the performance of each individual component used in our approach in Table 6. The results of HR Based FET is much better than the other two components. Also, its performance on BBN is much better than on FIGER (GOLD). This is because FIGER (GOLD) requires multi-class multi-label annotation, which means an entity can be annotated as two or more different type paths (e.g., annotated as both */organization/educational_institution*

and */location*), but BBN only annotates each entity with one type path (e.g., just /OR-GANIZATION/EDUCATIONAL). As a result, there are more errors while mapping the extracted hypernyms to FIGER (GOLD) types.

## 8. Conclusions

To address the problem that existing neural FET models may fail to make use of some beneficial semantic information, we propose to exploit semantic relations extracted from the sentence. We first build two individual components that performs FET based on hypernym relations and verb-argument relations respectively. Then, they are combined with a base neural model to make the final prediction. The framework of our approach makes it easy to combine more types of semantic relations. In our experiments, we find that both types of relations are helpful. We are able to improve the FET performance even when very strong base neural models are used.

## Acknowledgments

This paper was supported by the WeChat-HKUST WHAT Lab on Artificial Intelligence Technology, the Early Career Scheme (ECS, No. 26206717) and the Research Impact Fund (RIF, No. R6020-19) from the Research Grants Council (RGC) in Hong Kong.

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
