# OpenReview forum: "Exploiting Semantic Relations for Fine-grained Entity Typing"
_AKBC.ws/2020/Conference — AKBC 2020_

### Official Review · AnonReviewer1 · 2020-03-26
**straight-forward linguistic feature integration experiments**

**Rating:** 7
**Confidence:** 4

**Review:**

The paper describes an approach that models linguistic features extracted from the entity context and applies them to the fine-grained entity type (FET) prediction task. Experiments show that incorporating models for hypernym relation detection and semantic role labelling improve the performance.

* I would like to see more motivation for the FET task in the introduction. It is not clear why explicit type modelling is required for the down-stream tasks.

* There are many papers that report increase in performance on the NLP tasks, such as question answering, from incorporating these and other linguistic features that should be mentioned in the related work, e.g.

[1] Fabian Hommel, Philipp Cimiano, Matthias Orlikowski, Matthias Hartung: Extending Neural Question Answering with Linguistic Input Features. SemDeep@IJCAI 2019: 31-39
[2] Daniel Khashabi, Tushar Khot, Ashish Sabharwal, Dan Roth: Question Answering as Global Reasoning Over Semantic Abstractions. AAAI 2018: 1905-1914

* Semantic role labelling should be illustrated with examples and clearly motivated for the FET task.

* It is interesting to see dataset statistics with respect to the extracted features, e.g. how many hypernym mentions where detected, how many arguments for each of the roles in each of the datasets were extracted?

* Error analysis is missing. How many errors are propagated from the previous stages?

* "the hypernyms extracted by our approach are of high precision" What is the precision of hypernym extraction?

* Gating network architecture is not clearly specified. Is it the "MLP with two fully connected layers"? Formula 3 suggests a linear combination of vectors but the description above does not correspond to this formula.

* Abstract should contain more details on the datasets and results: "We conduct experiments on two commonly used datasets. The results show that our approach successfully improves fine-grained typing performance. "

---

> ### Author Response · Authors · 2020-04-08
> **Response to Reviewer 3**
>
> Thank the reviewer for the suggestions. We have revised the manuscript to address most of the concerns the reviewer has on our paper. But due to the 10-page space limit, we have to put most of the added content in the appendix.
>
> * I would like to see more motivation for the FET task in the introduction. It is not clear why explicit type modelling is required for the down-stream tasks.
>
> More detailed motivation for the FET task can be found in papers that propose this task and construct FET datasets (e.g., "Fine-Grained Entity Recognition" [Ling and Weld, 2012], which also includes experiments on applying FET to relation extraction and entity linking). Since the purpose of our work is to propose an approach, and AKBC only allows 10 pages, we chose not to include more motivation for the task, as it can be found in other related papers. We briefly explain how FET can help several down-stream tasks below.
>
> 1) FET helps knowledge base construction. FET recognize the fine-grained types of entities mentioned in texts, newly discovered type information can be added to a KB that is under construction. 2) FET helps relation extraction, because knowing the fine-grained types of two entities in a sentence makes it easier to infer their relation. E.g., if the types of two entities are /person/director and /person/film, then the relation is likely to be directorOf. 3) FET also helps Entity Linking, because knowing the type of the entity mention puts a constraint on the possible candidate entities. E.g., if we know that the type of a mention "USA" likely to be a sports_team, we can avoid linking it to the country.
>
> * There are many papers that report increase in performance on the NLP tasks, such as question answering, from incorporating these and other linguistic features that should be mentioned in the related work
>
> We have revised the manuscript so that the last paragraph of Related Work include such papers.
>
> * Semantic role labelling should be illustrated with examples and clearly motivated for the FET task.
>
> We revised the paper to include more explanation for the use of semantic role labeling in the first paragraph of Section 5 with examples. Appendix E also contains examples on how SRL helps FET. Generally speaking, when a mention matches an argument of a verb, we can sometimes infer its type from this verb or the combination of this verb and its other arguments.
>
> * It is interesting to see dataset statistics with respect to the extracted features, e.g. how many hypernym mentions where detected, how many arguments for each of the roles in each of the datasets were extracted?
>
> We added a section in the appendix (Appendix D) to include these statistics.
>
> * Error analysis is missing. How many errors are propagated from the previous stages?
>
> We added a section in the appendix (Appendix C) to report the performance of the three individual components. The performance of the SRL system on FIGER (GOLD) and BBN is hard to evaluate. The original paper that proposes the SRL system we use reports 88.1 F1 on CoNLL 05 and 86.2 F1 on CoNLL 12.
>
> * "the hypernyms extracted by our approach are of high precision" What is the precision of hypernym extraction?
>
> We manually annotate whether the hypernyms extracted for all the mentions in FIGER (GOLD) and 100 randomly selected mentions in BBN are correct. The precisions are 0.91 on FIGER (GOLD), 0.90 on BBN. The hypernyms that fully match the mention strings (e.g., the mention "drug" itself is regarded as a hypernym by our approach) are not included for annotation since although they are not correct hypernyms, they do not hurt the final performance (discussed in Appendix D).
>
> * Gating network architecture is not clearly specified. Is it the "MLP with two fully connected layers"? Formula 3 suggests a linear combination of vectors but the description above does not correspond to this formula.
>
> They are not inconsistent. The "MLP with two fully connected layers" outputs the weights ($w_1,w_2,w_3$) used in Formula 3.
>
> * Abstract should contain more details on the datasets and results: "We conduct experiments on two commonly used datasets. The results show that our approach successfully improves fine-grained typing performance."
>
> The revised version now contains more details.

---

> > ### Comment · AnonReviewer1 · 2020-04-17
> > **Revision accepted**
> >
> > I appreciate the effort authors took in addressing my comments that add more details to the manuscript and increase my score accordingly.

---

### Official Review · AnonReviewer2 · 2020-03-27
**semantic relations are helpful to fine-grained entity typing**

**Rating:** 8
**Confidence:** 5

**Review:**

The paper shows that semantic relations associated with mentions can be used to improve fine-grained entity typing. The whole model contains three parts: 1) Base FET Model 2) Hypernym Relation Model 3) Verb-argument Relation Model. Experimental results show that the integrated semantic relation information improves the final performance. The comparisons are extensive. The submission is well suited to the akbc conference.

---

> ### Author Response · Authors · 2020-04-08
> **Response to Reviewer 2**
>
> We thank the reviewer for the positive feedback. As mentioned in the paper, we will release code after the acceptance of this paper, so that people can use our approach easily.

---

### Official Review · AnonReviewer3 · 2020-03-30
**Well motivated idea with good improvement, although the method is a bit ad hoc**

**Rating:** 7
**Confidence:** 3

**Review:**

This work addresses fine-grained entity typing by leveraging semantic relations of the mention with other words in the same sentence. Specifically, the authors showed how to use hypernym relations and verb-argument relations. For the first, they used Wikidata to train a string match based candidate extraction and BERT-based verification model. For the second, they used an existing SRL system to extract relations between the verb and the mention. Then the two system each produce a prediction that is then combined with the base model through a gating network. The proposed method significantly improves over baselines. And they performed ablations studies to show the help from hypernym and verb-argument.

Strength:

1. The proposed approach is well motivated, and described clearly.

2. The advantage of the proposed modules (HR and VR) is validated through ablation studies.

Weakness:

1. The proposed method for combining leveraging different semantic relations is an ad hoc ensemble of separate systems. And each system has some other dependencies (extra data, for example, Wikidata, or external trained model, for example, AllenNLP SRL), which introduces more complexity in training.

2. It would help to show some examples to demonstrate the advantages of HR and VR. For example, what kind of sentences do they help and what kind of sentences do they hurt.

Questions:

Since the model combines three systems, I was wondering if the accuracy would drop, comparing to no HR or no VR, on sentences where there is no hypernym or no verb-argument structure detected. In other words, would adding HR or VR hurt performance on sentences where they only output zero vector?

---

> ### Author Response · Authors · 2020-04-08
> **Response to Reviewer 1**
>
> Thank the reviewer for the comments. Below are our responses:
>
> Weakness 1:
> It is true that our approach is more complicated than most end-to-end neural models. However, since we will release the code, the effort required to use our approach is not much. Our approach is more suitable for those that are willing to obtain better performance by using a slightly more complicated model.
>
> Weakness 2:
> We have revised the manuscript to include another section (Appendix E) in the appendix to discuss this. Several examples are listed in Table 5 and are discussed in Appendix E. In general, HR can help when the Base model is not able to recognize the true fine-grained type, but a correct hypernym is extracted and mapped to the target type set without mistake. However, HR may cause errors when the extracted hypernyms are incorrect, or are mapped to wrong types. VR can help when the verb or the combination of the verb and the other arguments helps the model to determine the correct type, but may also cause errors when the verb-(argument entity type) usage is uncommon or infrequent.
>
> Answer to the question:
> No, the accuracy would not drop on sentences where no hypernym or verb-argument relations are detected. This is because when we combine the results of the three models, we take a weighted sum of their predictions (See Equation 3). If no hypernym or verb-argument relations are detected in a sentence, $h_m$ and $g_m$ will be zero vectors, and thus has no effect to the final prediction vector $\hat{y}_m$.

---

### Decision · Program_Chairs · 2020-04-30

**Decision:**

Accept

**Comment:**

The paper make use of semantic relations (hypernym and verb-argument) to obtain the state of the art performance in entity typing, especially compared to strong baselines such as BERT. The paper presents an interesting message that linguistic features could be still important among the age of end-to-end methods. It is also clear that entity typing is crucial for constructing knowledge bases, making the paper quite appropriate for the proceedings of AKBC.